# A Deep Neural Network Regularization Measure: The Class-Based Decorrelation Method

Chenguang Zhang [1,*], Tian Liu [2] and Xuejiao Du [1]

1   School  of Mathematics and Statistics, Hainan University, Haikou 570100, China; xuejiao_du@hainanu.edu.cn
2   School of Information and Communication Engineering, Hainan University, Haikou 570100, China; liutian@hainanu.edu.cn
*   Correspondence: chenguang_zhang@hainanu.edu.cn

**Abstract:** In response to the challenge of overfitting, which may lead to a decline in network generalization performance, this paper proposes a new regularization technique, called the class-based decorrelation method (CDM). Specifically, this method views the neurons in a specific hidden layer as base learners, and aims to boost network generalization as well as model accuracy by minimizing the correlation among individual base learners while simultaneously maximizing their class-conditional correlation. Intuitively, CDM not only promotes diversity among the hidden neurons, but also enhances their cohesiveness among them when processing samples from the same class. Comparative experiments conducted on various datasets using deep models demonstrate that CDM effectively reduces overfitting and improves classification performance.

**Keywords:** deep neural network; generalization ability; regularization method

## 1. Introduction

Deep neural networks (DNNs) have demonstrated the highest accuracy in many artificial intelligence (AI) tasks, including computer vision, speech recognition, and robotics, due to their deep learning architecture and parallel learning mechanism, which guarantees its powerful expressive capacity to learn complex things [1]. However, overparameterization may also bring the risk of overfitting. How to maintain or even improve the learning advantages brought by deep architecture while avoiding overfitting is a continuous research topic in the field of deep learning.

In recent years, some studies have shown that DNNs may have a self-regularization effect [2], and the classical generalization theory [3,4] may not be applicable to deep learning. They suggested that the mechanism and effect of the regularization methods such as $L_2$ regularization [5,6], Weight Decay [7], Dropout [8], etc., which enhance the generalization ability by limiting the model complexity, need to be further explored and clarified.

On the other hand, certain studies also argue that traditional mean deviation theory remains effective, but requires a revision for the concept of model complexity, which should contain not only the parameter numbers but also the data, optimization methods, and model structure [9,10]. Some corresponding explicit regularization methods are explored and testified to have positive effect on reducing the generalization gap (performance gap between the test dataset and the train dataset) [11–13]. For example, Achille et al. [14] proposed that the mutual information between labels and model parameters can be used as a regularization term to predict sharp phase transitions between underfitting and overfitting of random labels.

Recently, in contrast to treating the whole deep learning network as a single black box, several studies begin to explore its parallel mechanism. By viewing the network output as a collection of fundamental mappings or combinations of individual learners [12,15,16], these studies analyze the influence of their correlations on the generalization capabilities of

deep learning models, introducing novel concepts for devising regularization techniques to enhance generalization. This concept of correlation is referred to as the disentanglement measure in representation learning and the diversity measure in ensemble learning. Several well-established disentanglement measures [17,18] as well as diversity measures [19,20] have been introduced, and their validity has been verified theoretically and experimentally.

Inspired by ensemble learning and representation learning, this paper proposes a novel approach based on class labels for decorrelating neurons within a hidden layer of deep neural networks, called the class-based decorrelation method (CDM). This method takes into account both the unsupervised decorrelation between hidden neurons and the supervised requirement for collaboration of neurons in the same class. The unsupervised decorrelation term has been explored in a large amount of the literature. For example, Cogswell et al. [11] and Gu et al. [12] consider hidden layer neurons, or their groupings, as basic mappings to improve the generalization of deep networks by limiting the correlation between mappings. In the field of representation learning, methods such as variational autoencoders (VAE) [21] and $\beta$-VAE [22] have demonstrated the beneficial impact of disentanglement on downstream tasks, including improving interpretability, controllability, and robustness.

Furthermore, the research conducted by Zhou et al. [23] in the field of ensemble learning has shown that in addition to the aforementioned unsupervised term, a supervised term formalized as mutual information that reflects clustering characteristics is also helpful in improving prediction accuracy on future data. A similar conclusion was drawn by Zhang et al. [24], who decomposed the upper bound of the generalization error from information perspective. However, considering the complexity of estimating mutual information and that covariance is straightforward proxy for mutual information, CDM provides a regularization method with the above supervised and unsupervised terms estimated by covariance. Intuitively, CDM encourages the diversity among hidden neurons to diminish data redundancy while simultaneously strengthening neuron collaboration when dealing with samples from the same class, which helps to preserve useful information for classification, ensuring significantly higher classification accuracy and improved generalization performance.

We conducted comparative experiments on different depths of network structures. A large number of experiments have shown that compared to other regularization methods, CDM can effectively reduce overfitting and generalization errors while improving model accuracy. Due to the simplicity of CDM, it can be easily applied to any layer of a neural network. In addition, the experiment also verified that the class label-related terms in the supervised setting are a key factor in improving the generalization ability of DNNs.

## 2. Related Work

Understanding the generalization mechanism of deep learning and how it differs from classic methods is a crucial step to improve the generality of deep models. Recently, some studies have sought to provide more compact new generalization upper bounds by taking the special situations of the current training dataset, model frameworks, training methods, etc., into account. Here are some notable examples. Bartlett and Maiorov [25], Bartlett and Harvey [26], and Yang et al. [27] proposed upper bounds of DNNs based on the VC-dimension. Dziugaite and Roy [28] and Neyshabur et al. [29,30] used PAC-Bayesian analysis to derive bounds related to deep polynomials. Neyshabur et al. [31] proved the depth exponential dependence bound of ReLU networks by using Rademacher complexity, and Bartlett et al. [32] proposed a norm-based generalization bound of neural networks. Based on a ReLU kernel function, Arora et al. [33] proposed a generalization error bound for two-layer ReLU networks with fixed second-layer weights. Daniely and Granot [34] obtained improved bounds for constant-depth fully connected networks. They also introduce some empirical-based measures, such as the Fisher–Rao norm proposed by Liang et al. [35] and the study of generalization gaps with various dependencies, including distance from initialization, conducted by Jiang et al. [36].

The parallel mechanism of neural networks naturally lends itself to mutual inspiration from ensemble learning. Moreover, the divergence measure among individual learners in ensemble learning usually corresponds to the degree of disentanglement in representation learning. For example, the variational autoencoder (VAE) [21] applies a constraint on the learned representations of a neural network based on the Kullback–Leibler (KL) divergence between the variational posterior distribution and the true posterior distribution, thereby endowing the method with the potential for disentanglement capability. Furthermore, methods like $\beta$-VAE [22], DIP-VAE [37], and $\beta$-TCVAE [38] enhance the disentanglement ability and improve model generalization by incorporating implicit or explicit inductive biases into the original VAE loss function. Although the above method successfully implements the attribute "encouraged" by the corresponding loss, some researchers argue that identifying a well-disentangled model is virtually impossible without inductive biases in the absence of supervised settings.

Another important paradigm for improving the generalization performance of neural networks is regularization. Traditional regularization methods focus on constraining the complexity of the model, such as Weight Decay [7], Dropout [8] and DropConnect [39]. Although these methods have made improvements in terms of generalization, the random reduction in parameters may affect the expressive power of the model. In response, some research focuses on improving the generalization of neural networks while preserving their expressive power. For example, Bengio and Bergstra [40] proposed a pretraining algorithm for learning decorrelation features. Bao et al. [41] discussed decorrelation activation using incoherent training. Combining ensemble learning, Gu et al. [12] and Yao et al. [13] proposed corresponding decorrelation regularization methods by treating the network output as a combination of learners. Ayinde et al. [20] proposed an effective method for regularizing deep neural networks by leveraging the correlation between features. However, these regularization methods only emphasize the impact of unsupervised decoupling in hidden neurons, while neglecting the positive effect of supervised term that captures clustering characteristics on DNNs. Addressing this, Zhang et al. [24] obtained a regularization method containing both supervised and unsupervised items from the perspective of mutual information, and emphasized the importance of label-based inductive bias under the supervision setting to improve the network generalization ability. While this method is theoretically feasible, it may be less accessible due to the difficulty in estimating mutual information. Our work aims to overcome these limitations and proposes a simpler and more generalizable solution.

## 3. Class-Based Decorrelation Method

This section provides a detailed introduction to the class-based decorrelation method (CDM), which is a regularization method aimed at promoting the diversity of hidden neurons while enhancing collaboration among neurons when processing samples from the same class.

### 3.1. Theoretical Motivation

CDM is grounded in the newly introduced label-based diversity measure, which amalgamates the unsupervised diversity measure with supervised class-conditional diversity measure among hidden neurons, shedding light on the improvement of generality without compromising classification accuracy. Let $S = \{(X, Y) \mid (X, Y) \in \mathcal{X} \times \mathcal{Y}\}$ be the given dataset, and let $h_i(X)$ represent the activation of the sample in the $i$-th hidden neuron; the label-based diversity (LDiversity) among hidden neurons can be expressed as:

$$D_{LB}(S) = I(h_1(X); \ldots; h_m(X)) - I(h_1(X); \ldots; h_m(X) \mid Y), \tag{1}$$

where the initial mutual information term in the right-hand side serves the purpose of extracting independent features, aligning with the typical notion of an unsupervised diversity measure. Conversely, the second class-conditional term is intricately linked

with labels, reflecting the underlying inductive bias inherent in the new representations of samples.

The work of Zhou et al. [23] shows that if only 0–1 loss is permitted, given $\hat{h} = (h_1, h_2, \ldots, h_m)$ as a set of base classifiers $h_i$ for the labeled sample $(X, Y)$, and $C(\cdot)$ as any given combination function that minimizes the probability $P(C(\hat{h}(X)) \neq Y)$, the probability is bounded as follows:

$$P(C(\hat{h}(X)) \neq Y) \leq \frac{H(Y) - \sum_{i=1}^{m} I(h_i(X); Y) + D_{LB}(S)}{2}, \tag{2}$$

which indicates that there is an inverse relationship between the test error and LDiversity.

Zhang et al. [24] obtained a similar result. Further, their work is directly founded on the decomposition of the generalization error bound and reveals that reducing the value of LDiversity can enhance the generality.

Given the intricacies in estimating mutual information and the direct correlation between covariance and mutual information where mutual information is a monotone transformation of covariance, we opt to use covariance as a proxy in Equation (1), thereby deriving the following expression:

$$
\begin{aligned}
D'_{LB}(S) = &\sum_{i,j} \mathbb{E}_S\left[(h_i(X) - \mathbb{E}_S[h_i(X)])(h_j(X) - \mathbb{E}_S[h_j(X)])\right] \\
&- \sum_{i,j} \mathbb{E}_{S_Y}\left[(h_i(X) - \mathbb{E}_{S_Y}[h_i(X)])(h_j(X) - \mathbb{E}_{S_Y}[h_j(X)]) \mid Y\right].
\end{aligned}
\tag{3}
$$

Clearly, the covariance-based redefinition of LDiversity still retains the core idea of Equation (1), with the goal of encouraging diversity among hidden neurons while fostering stronger collaboration among neurons within the same class. Furthermore, covariance exhibits a straightforward and easily estimable characteristic, making CDM a more versatile and readily implementable approach.

### *3.2. CDM Application in the Fully Connected Layer*

As illustrated in Figure 1, when applying CDM to a specific fully connected layer of a neural network, each hidden neuron within this layer is treated as a base learner. By exemplifying the calculation on a batch of samples, we demonstrate the specific process of applying CDM to any fully connected layer.

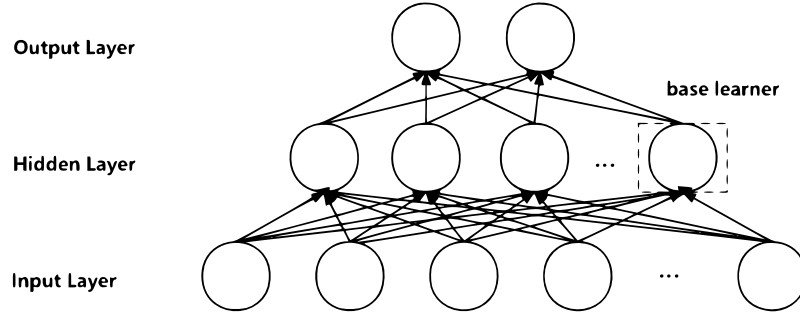

**Figure 1.** Architecture of a fully connected neural network based on CDM. Each neuron of the specified hidden layer is treated as a base learner.

Let $n$ be the number of neurons in the target hidden layer. Consider a batch of $N$ samples categorized into $K$ classes, where each class contains $n_k$ samples (with $k = 1, \ldots, K$) and $N = \sum_{k=1}^{K} n_k$. Let $h_i^n$ represent the activation originating from the $n$-th sample in the input to the $i$-th hidden neuron in the target layer. To represent the correlation between the $i$-th and $j$-th base learners across the entirety of the mixed distribution, we introduce the notation $LnCor_{ij}^{mix}$, which is formally expressed as follows:

$$LnCor_{ij}^{mix} = \frac{1}{N} \sum_{n=1}^{N} (h_i^n - \mu_i)\left(h_j^n - \mu_j\right), \tag{4}$$

$$\mu_i = \frac{1}{N} \sum_{n=1}^{N} h_i^n, \tag{5}$$

where $\mu_i$ denotes the activation mean of the $i$-th base learner across $N$ samples. As covariance quantifies the correlation between variables, lower covariance usually implies reduced correlation. Consequently, we need to minimize this item, aiming to maintain a minimal level of correlation among the base learners, thus fostering greater diversity in sample representations.

The class-conditional correlation between the $i$-th and $j$-th base learners denoted as $LnCor_{ij}^{class}$ is expressed as follows:

$$LnCor_{ij}^{class} = \sum_{k=1}^{K} \frac{n_k}{N} LnCor_{(k,ij)}^{class}, \tag{6}$$

$$LnCor_{(k,ij)}^{class} = \frac{1}{n_k} \sum_{n=1}^{n_k} \left(h_{(k,i)}^n - \mu_i^k\right)\left(h_{(k,j)}^n - \mu_j^k\right), \tag{7}$$

$$\mu_i^k = \frac{1}{n_k} \sum_{n=1}^{n_k} h_{(k,i)}^n, (k = 1, \dots, K), \tag{8}$$

where $LnCor_{(k,ij)}^{class}$ represents the covariance between the $i$-th and $j$-th base learners within the $k$-th class; $h_{(k,i)}^n$ represents the activation value from $n$-th sample in the $k$ class to the $i$-th hidden neuron, and $\mu_i^k$ signifies the mean activation value of the $i$-th hidden neuron over the samples within this class. Formula (4) pertains to the class labels and describes local clustering features captured by hidden neurons. Enhancing the class-conditional term, especially by maximizing its value, is anticipated to promote cohesiveness of base learners when the input samples originate from the same class.

By utilizing $LnCor_{ij}^{mix}$ and $LnCor_{(k,ij)}^{class}$ as the entries in the $i$-th row and $j$-th column of matrices, we can construct the covariance matrix $LnCor^{mix}$ and the class-conditional covariance matrix $LnCor^{class}$, respectively. Let $S_L^{mix}$ and $S_L^{class}$ denote the final total correlation over the mixed distribution and total class-conditional correlation, respectively. From these, we derive the ultimate penalty regularization term, denoted as $LnCor_{Loss}$:

$$LnCor_{Loss} = ln(S_L^{mix}) - ln(S_L^{class}), \tag{9}$$

$$S_L^{mix} = \frac{1}{2}(\|LnCor^{mix}\|_F^2 - \|diag(LnCor^{mix})\|_2^2), \tag{10}$$

$$S_L^{class} = \frac{1}{2}(\|LnCor^{class}\|_F^2 - \|diag(LnCor^{class})\|_2^2), \tag{11}$$

where $\|\cdot\|_F$ is the Frobenius norm, the $diag(\cdot)$ operator returns the main diagonal elements of the matrix as vectors, and $ln(\cdot)$ represents logarithmic operation. It is worth noting that the diagonal elements of the two matrices have been excluded from their respective correlation measures since these elements indicate self-correlation. Minimizing the regularization term $LnCor_{Loss}$ is expected to encourage nonredundant representations of samples.

Finally, the total loss of the neural network applying CDM at the fully connected layer can be expressed as:

$$T_{Loss} = E_{Loss} + \lambda LnCor_{Loss}, \tag{12}$$

where $E_{Loss}$ is the cross-entropy loss, and $\lambda \geq 0$ is the hyperparameter.

Specially, to illustrate the training process on the $l$-th layer, we assume that $W^l$ represents the weight matrix from the $(l-1)$-th to the $l$-th layer, and $H^l$ denotes the activations of the $l$-th layer. Now, let us examine the gradient of the total loss for a specific sample:

$$\frac{\partial T_{Loss}}{\partial W_i^l} = \frac{\partial T_{Loss}}{\partial h_i} \cdot \frac{\partial h_i}{\partial W_i^l}, \tag{13}$$

$$\frac{\partial LnCor_{Loss}}{\partial h_i} = \frac{1}{N} \sum_{i \neq j} LnCor_{i,j} \cdot (h_j - \mu_j) + \sum_k \frac{n_k}{N} \sum_{i \neq j} LnCor_{i,j} \cdot \left(h_j^k - \mu_j^k\right), \tag{14}$$

$$\frac{\partial h_i}{\partial W_i^l} = H^{l-1}, \tag{15}$$

where we have not shown the gradient of the cross-entropy part since it is common. From Equations (13)–(15), it has been demonstrated that the gradient can be conveniently computed through covariance-based backpropagation. Then, the obtained gradient is subsequently utilized for updating the weights.

*3.3. CDM Application in Convolutional Layer*

The application of CDM in convolutional layers differs from its use in fully connected layers in that each neuron is no longer treated as an individual base learner; instead, the convolutional feature map is considered as the base learner. Two primary reasons underlie this approach: (1) Decorrelating all the hidden neurons within a convolutional layer incurs a significant computational cost. (2) Distinct neurons may share the same convolutional feature map, rendering decorrelation efforts on these neurons essentially futile. In fact, it is the feature map that fundamentally reflects the mapping relationship from samples to their representations. In a similar vein, we illustrate how CDM can be applied to any convolutional layer and trained on a batch of samples with size $N$.

Given the convolutional layer to be processed, let $M$, $H \cdot W$ be the number of convolutional feature maps and the output spatial dimensions. We use $CoCor_{ij}^{mix}$ to represent the correlation between the $i$-th and $j$-th base learners over the mixed distribution, which is expressed as follows:

$$CoCor_{ij}^{mix} = \frac{1}{N} \sum_{n=1}^{N} (g_i^n - \overline{\mu}_i)\left(g_j^n - \overline{\mu}_j\right), \tag{16}$$

$$\overline{\mu}_i = \frac{1}{N} \sum_{n=1}^{N} g_i^n, \tag{17}$$

$$g_i^n = \frac{1}{HW} \sum_{h=1}^{H} \sum_{w=1}^{W} v_i^{(h,w)}, \tag{18}$$

where $v_i^{(h,w)}$ represents the value whose coordinate position is $(h, w)$ in the output of the $i$-th convolutional feature map; $g_i^n$ is regarded as the activation originated from the $n$-th sample in the input batch of the $i$-th base learner, and $\overline{\mu}_i$ represents the mean of the activations across the samples. Correspondingly, the conditional correlation between the $i$-th and $j$-th base learner, denoted as $CoCor_{ij}^{class}$, is defined as follows:

$$CoCor_{ij}^{class} = \sum_{k=1}^{K} \frac{n_k}{N} CoCor_{(k,ij)}^{class}, \tag{19}$$

$$CoCor_{(k,ij)}^{class} = \frac{1}{n_k} \sum_{n=1}^{n_k} \left(g_{(k,i)}^n - \overline{\mu}_i^k\right)\left(g_{(k,j)}^n - \overline{\mu}_j^k\right), \tag{20}$$

$$\overline{\mu}_i^k = \frac{1}{n_k} \sum_{n=1}^{n_k} h_{(k,i)}^n, (k = 1, \dots, K), \tag{21}$$

$$g_{(k,i)}^n = \frac{1}{HW} \sum_{h=1}^{H} \sum_{w=1}^{W} v_{(k,i)}^{(h,w)}, \tag{22}$$

where $CoCor_{(k,ij)}^{class}$ represents the covariance between the $i$-th and $j$-th feature maps within the class $k$; $g_{(k,i)}^n$ denotes the activation value of the $n$-th sample in $k$-th class by the $i$-th hidden neuron, and $\overline{\mu}_i^k$ signifies the mean activation value across different samples. Similar to the fully connected layer, we construct the covariance matrix $CoCor^{mix}$ and the class conditional covariance matrix $CoCor^{class}$ by using $CoCor_{ij}$ and $CoCor_{(k,ij)}^{class}$ as entries. The regularization term $CoCor_{Loss}$ for the convolutional layer is then defined as follows:

$$CoCor_{Loss} = ln(S_C^{mix}) - ln(S_C^{class}), \tag{23}$$

$$S_C^{mix} = \frac{1}{2}(\|LnCor^{mix}\|_F^2 - \|diag(LnCor^{mix})\|_2^2), \tag{24}$$

$$S_C^{class} = \frac{1}{2}(\|LnCor^{class}\|_F^2 - \|diag(LnCor^{class})\|_2^2). \tag{25}$$

where $S_C^{mix}$ and $S_C^{class}$ are the total correlation over the entire mixed distribution and the total class-conditional correlation.

Adding $CoCor_{Loss}$ to the cross-entropy loss, we define the final loss for neural networks with convolutional layer as follows:

$$T_{Loss} = E_{Loss} + \gamma CoCor_{Loss}, \tag{26}$$

where $\gamma \geq 0$ is the hyperparameter.

## 4. Experimental Section

In this section, to validate the efficacy of CDM in deep neural networks, we performed experiments on diverse datasets using different neural networks of varying depths. The control group encompassed several methods: nonregularization (None); Dropout [8] with a random probability of 0.5; the decorrelation regularization method (DeCov) [11] with a hyperparameter set to 0.1; and the ensemble-based decorrelation method (EDM) [12] with a balance parameter of 0.1. Notably, since DeCov, EDM, and CDM all fall under decorrelation techniques, for fairness, they were applied to the same hidden layer. For Dropout, it is commonly utilized for fully connected layers.

Additionally, we also tested the effectiveness of CDM when applied to the state-of-the-art methods, including Inception [42] and MobileNet [43], focusing on the challenge of relatively limited training samples encountered commonly in real-world applications.

### 4.1. Datasets

The study presented in this paper makes use of two publicly available datasets: MNIST [44], CIFAR-10 [45] and mini-ImageNet [46]. A brief overview of these datasets is presented below.

- **MNIST:** The MNIST dataset is a collection of handwritten digit images, totaling 70,000 grayscale images across 10 different classes. Each image is 28 pixels in height and 28 pixels in width. The training set comprises 60,000 images, while the test set contains 10,000 images.
- **CIFAR-10:** The CIFAR-10 dataset comprises 10 classes, with each class containing 6000 color images sized at $32 \times 32$ pixels and composed of RGB three-channel data. The training set encompasses 50,000 images, while the test set includes 10,000 images.

- **Mini-ImageNet**: The mini-ImageNet dataset consists of 50,000 training images and 10,000 testing images, evenly distributed across 100 classes. The images have a size of $84 \times 84 \times 3$.

Moreover, to effectively highlight the generalization capabilities of each method, we introduced noise to the test images. The noise adheres to a standard Gaussian distribution. For MNIST and CIFAR-10 as well as mini-ImageNet, we multiplied the noise by weights of 0.2 and 0.05, respectively, when adding it to the original pixel values. It is important to note that prior to introducing the noise, we standardized pixel values within the range $[-1, 1]$ (refer to Figure 2).

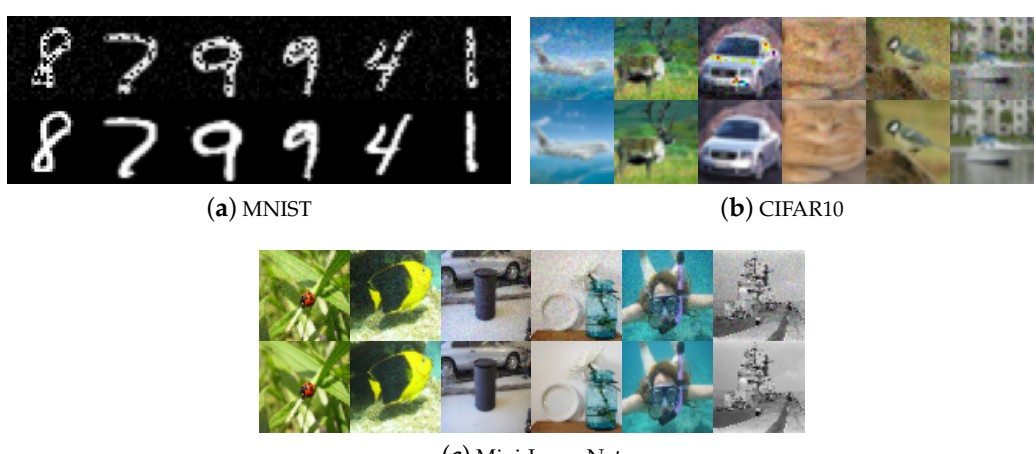

(**a**) MNIST        (**b**) CIFAR10

(**c**) Mini-ImageNet

**Figure 2.** Some examples of introducing Gaussian noise to (**a**) MNIST, (**b**) CIFAR-10 and (**c**) Mini-ImageNet datasets to generate noisy images (**top row**) from original images (**bottom row**).

### 4.2. Network Frameworks

The study employs fully connected neural networks (FCNNs) and residual neural networks (ResNets) with two distinct depths. In this context, "depth" refers to the number of layers that undergo parameter updates during training, encompassing convolutional layers, fully connected layers, and so forth. Additionally, aiming at the challenge of relatively limited training samples, the learning architectures from approaches such as Inception and MobileNet are included for comparative analysis.

The architecture of FCNNs is composed of layers in the order of D(512)-D(256)-D* (100)-D(256)-D(128)-D(64), where D($N$) signifies a dense layer with $N$ neurons. The introduced regularizer is specifically applied to the D* layer. Furthermore, for residual neural networks, we make use of ResNet18 [47] and ResNet50 [48]. Specific details regarding the parameters can be found in Table 1. Besides ResNet18 and ResNet50, the learning methods of Inception and MobileNet are also utilized. All the models are pretrained on the ImageNet dataset [49]. Throughout the training phase, transfer learning and fine-tuning methodologies are employed, allowing for active adjustment of the model's layer parameters without freezing. This approach ensures continuous updates to the parameters to enhance the network's performance. Notably, the regularizer is implemented either on the topmost fully connected layer (excluding the output layer) or on the convolutional layer, as applicable. The entire experiment is carried out using PyTorch [50].

**Table 1.** Structure diagram of residual neural networks at different depths.

| Layer | 18-Layer (ResNet18) | 50-Layer (ResNet50) |
|:---:|:---:|:---:|
| Conv 1 | $7 \times 7, 64$, stride2 | |
| | $3 \times 3$, maxpool, stride2 | |
| Conv 2 | $\begin{bmatrix} 3 \times 3, 64 \\ 3 \times 3, 64 \end{bmatrix} \times 2$ | $\begin{bmatrix} 1 \times 1, 64 \\ 3 \times 3, 64 \\ 1 \times 1, 256 \end{bmatrix} \times 3$ |
| Conv3 | $\begin{bmatrix} 3 \times 3, 128 \\ 3 \times 3, 128 \end{bmatrix} \times 2$ | $\begin{bmatrix} 1 \times 1, 128 \\ 3 \times 3, 128 \\ 1 \times 1, 512 \end{bmatrix} \times 4$ |
| Conv4 | $\begin{bmatrix} 3 \times 3, 256 \\ 3 \times 3, 256 \end{bmatrix} \times 2$ | $\begin{bmatrix} 1 \times 1, 256 \\ 3 \times 3, 256 \\ 1 \times 1, 1024 \end{bmatrix} \times 6$ |
| Conv5 | $\begin{bmatrix} 3 \times 3, 512 \\ 3 \times 3, 512 \end{bmatrix} \times 2$ | $\begin{bmatrix} 1 \times 1, 512 \\ 3 \times 3, 512 \\ 1 \times 1, 2048 \end{bmatrix} \times 3$ |
| Last | average pool/D(1000), D(10)/D(100), softmax | |

The structure of residual neural network with different depths (18, 50) is given in the table. The number of neurons in the output layer is aligned with the quantity of categories present in the datasets.

### 4.3. Effect on the Covariance Gap

We initially conducted an investigation to determine whether the CDM can boost the diversity among hidden neurons while simultaneously strengthening neuron collaboration when the samples are from the same classes. We varied the weight $\lambda$ of the regularization term in Equations (12) and (26) from 0 to 0.4 in increments of 0.1. The average results of experiments conducted on the MNIST and CIFAR-10 datasets are illustrated in Figure 3, where FCNNs and ResNet50 with the regularizer applied on their topmost convolutional layers were employed for the above datasets, respectively.

Significantly, the covariance gap, denoting the variance between the covariance and the class-conditional covariance, displays a decreasing pattern as the number of iterations rises when $\lambda > 0$. This trend indicates that CDM actively aids in decreasing regularizer values, aligning with its intended design. Notably, with $\lambda = 0$, the regularizer value remains static, consistently near zero but higher compared to other instances. Moreover, an increase in $\lambda$ value correlates with a reduction in the regularizer over the same training period.

### 4.4. Hyperparameter

To assess the impact of hyperparameters on experimental outcomes, we conducted multiple experiments using exclusively FCNNs for the MNIST dataset and ResNet50 for the CIFAR-10 dataset. Specifically, for the CIFAR-10 dataset, the regularizer was implemented on the topmost convolutional layer of ResNet50. The test results on the original test set, which is devoid of artificial noise, are illustrated in Figure 4. Upon observing the trends portrayed in Figure 4, it becomes apparent that a gradual increment in the regularization term weight within a narrow range leads to improved classification accuracy and a slight reduction in generalization error. However, if the weight is larger, the classification performance may, conversely, begin to slightly decrease. Nonetheless, the experimental findings suggest that the influence of hyperparameters on performance remains relatively stable, providing a broad spectrum of choices. Given that CDM exhibits a relatively minor covariance gap and performs quite well at $\lambda = 0.3$, subsequent experiments will maintain a fixed $\lambda$ at 0.3.

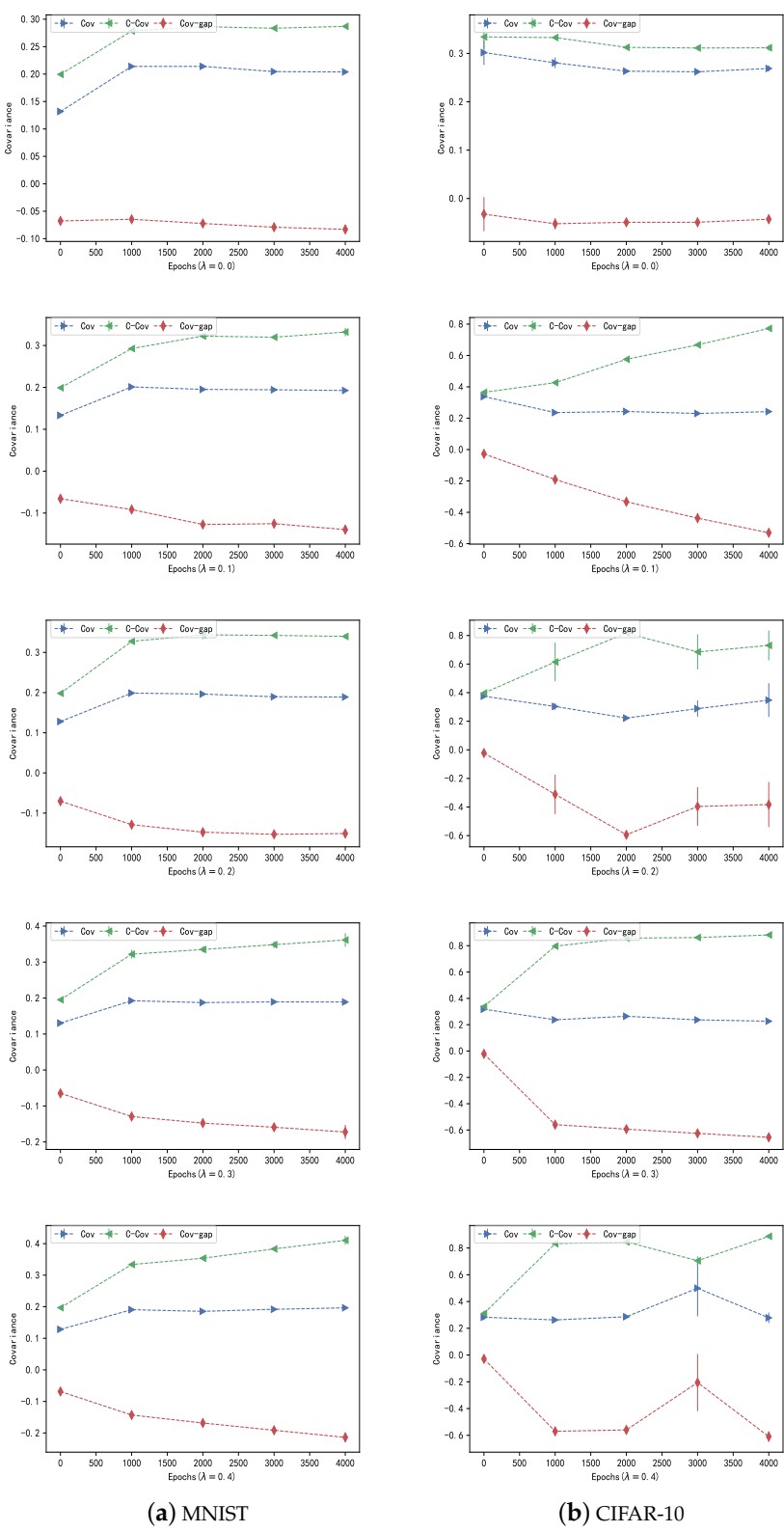

**Figure 3.** Given the values of $\lambda$ in the range $[0, 0.4]$, the changes in covariance (Cov), class-conditional covariance (C-Cov), and the covariance gap (Cov-gap) with increasing training iterations on (**a**) MNIST and (**b**) CIFAR-10 datasets.

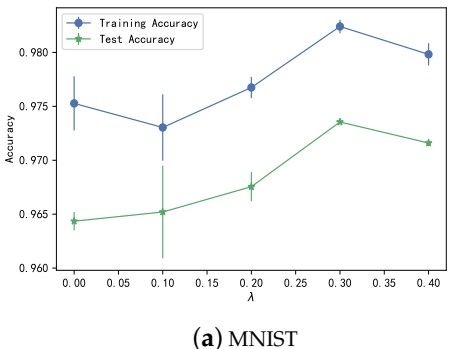 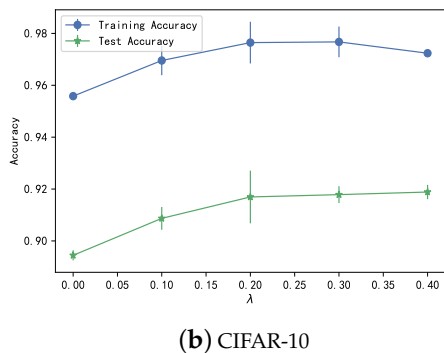

| (**a**) MNIST | (**b**) CIFAR-10 |

**Figure 4.** The changes in classification accuracy by varying the value of $\lambda$ on (**a**) MNIST and (**b**) CIFAR-10 datasets.

### 4.5. Experiment on Fully Connected Layer

The purpose of this experiment is to evaluate the effect of CDM in contrast to other regularization techniques when applied to the fully connected layer. Throughout all method experiments, the Adam algorithm was utilized for training, the initial learning rate was set at 0.001, and a consistent batch size of 128 was maintained. Comparative experiments were conducted on both the MNIST and CIFAR-10 datasets using images with added noise. The final results of the experiment were averaged with the accuracy of five experiments and are recorded in Table 2.

The experimental results, as presented in Table 2, demonstrate that CDM achieves significant test accuracy and the smallest train–test accuracy gap on both the MNIST and CIFAR-10 datasets. Specifically, in the case of MNIST, CDM yielded an impressive 1.8 improvement in test accuracy and simultaneously reduced the train–test accuracy gap by 0.8 when contrasted to suboptimal methods. Similarly, for CIFAR-10, CDM exhibits a remarkable 2.6 increase in test accuracy, accompanied by a substantial reduction of 1.9 in the train–test accuracy gap. It is noteworthy that CDM's impact on network generalization becomes progressively more pronounced as the dataset's complexity increases.

Furthermore, it is pertinent to mention that both DeCov and CDM share the same unsupervised disentanglement term. However, DeCov as well as EDM do not consistently enhance performance. For example, on the MNIST dataset, their test accuracy is marginally lower compared to methods with no regularization. This could be due to the dataset's simplicity, where mandating independence among hidden neurons disrupts their intrinsic relationships. Therefore, attributing the superior performance of CDM to the role of the supervised term is a reasonable and well-supported conclusion, indicating that this supervision term can help extract beneficial clustering information for classification.

**Table 2.** Compar tive experiment of different datasets on the full connection layer.

| Method | MNIST | | | CIFAR-10 | | |
|---|---|---|---|---|---|---|
| | Train | Test | Train–Test | Train | Test | Train–Test |
| None | 97.52 ±0.14 | 92.10 ± 0.67 | 5.42 | 95.58 ± 0.33 | 71.33 ± 1.48 | 24.25 |
| Dropout | 97.58 ± 0.06 | 91.41 ± 0.58 | 6.17 | 96.99 ± 0.35 | 75.20 ± 0.40 | 21.78 |
| DeCov | 97.78 ± 0.10 | 91.35 ± 1.11 | 6.42 | 95.81 ± 0.16 | 74.35 | 21.42 |
| EDM | 98.18 ± 0.05 | 91.95 ± 0.07 | 6.23 | 96.26 ± 0.27 | 75.23 ± 1.01 | 21.03 |
| CDM | 98.24 ± 0.12 | **93.99 ± 0.16** | **4.25** | 96.95 ± 0.23 | **77.83 ± 0.30** | **19.12** |

The average accuracy of five repeated tests was used as the final result. Best scores are shown in bold.

### 4.6. Experiments on Convolutional Layer

We applied CDM, DeCov, and EDM to the final convolutional layer of the network, comparing CDM with other regularization methods in terms of network performance. The training setups, such as learning rate and batch size, remained consistent with those

used for fully connected layers. We explored different depths of residual neural networks on the CIFAR-10 dataset with added artificial noise. The results, specifically the average classification accuracy derived from five experiments, are documented in Table 3.

As shown in Table 3, CDM consistently achieved the highest test accuracy and the smallest train–test accuracy gap in all comparative experiments, affirming its effective impact on enhancing network generalization, particularly for complex datasets. Moreover, it is worth noting that the overall performance of all regularization methods based on ResNet18 appears more similar compared to those using ResNet50, which could be attributed to the ResNet18 model's simplicity and reduced susceptibility to overfitting, leading to a relatively constrained impact of regularization. Moreover, at times, the performance of Decov and EDM falls even below that of the Dropout method, especially when confronted with more intricate network structures. Given that the regularization terms in Decov and EDM exclusively encompass unsupervised elements, the superior performance of CDM can be credited to its incorporation of a supervised term, thereby preventing the disruption of valuable classification information.

**Table 3.** Comparative experiment of different methods on CIFAR-10 dataset with regularizers applied on the topmost convolutional layer of ResNet18 and ResNet50.

| Method | ResNet18 | | | ResNet50 | | | |
|---|---|---|---|---|---|---|---|
| | Train | Test | Train–Test | Train | Test | Train–Test | |
| None | $97.12 \pm 0.04$ | $76.94 \pm 0.07$ | 20.17 | $97.10 \pm 0.04$ | $78.7 \pm 0.09$ | 18.40 | |
| Dropout | $97.22 \pm 0.02$ | $80.05 \pm 0.19$ | 17.16 | $97.62 \pm 0.02$ | $81.65 \pm 0.08$ | 15.96 | |
| DeCov | $97.34 \pm 0.09$ | $80.19 \pm 0.12$ | 17.15 | 96.81 | $\pm 0.06$ | $79.98 \pm 0.10$ | 16.82 |
| EDM | $97.17 \pm 0.07$ | $79.99 \pm 0.11$ | 17.18 | $96.94 \pm 0.05$ | $80.12 \pm 0.10$ | 16.81 | |
| CDM | $97.32 \pm 0.04$ | **$81.14 \pm 0.09$** | **16.17** | $97.09 \pm 0.06$ | **$82.23 \pm 0.08$** | **14.85** | |

The average accuracy of five repeated tests was used as the final result. Best scores are shown in bold.

### 4.7. Experiments with a Comparatively Small Training Dataset

In practical applications, the relative scarcity of training samples is a commonly encountered issue, especially in the case of deep network structures, which can lead to problems related to overfitting. Specifically, the classification task on the mini-ImageNet dataset exemplifies such a challenge, as each class among the 100 classes comprises only 500 image samples for training. This characteristic renders learning methods applied to this dataset particularly susceptible to overfitting.

To further assess the effectiveness of CDM in addressing this challenge, we applied it to pretrained Inception and MobileNet models, both of which have demonstrated state-of-the-art (SOTA) results on the ImageNet dataset. A comprehensive comparative analysis of experimental results was conducted both before and after the addition of CDM on mini-ImageNet. Additionally, noise was introduced into the test images. As depicted in Table 4, it is evident that all methods exhibit a certain degree of overfitting. However, the methods incorporating CDM result in improved classification accuracy for all methods compared to the original approach, with increases of 1.7 and 4.72 points, respectively. Concurrently, the integration of CDM reduces generalization errors by 3.53 and 2.31 points, respectively. The comparison with the SOTA methods strongly demonstrates the effectiveness of CDM regularization.

**Table 4.** Applying CDM to the topmost layer of Inception and MobileNet methods: a performance comparison before and after CDM implementation.

| Method | Inception | | | MobileNet | | |
|---|---|---|---|---|---|---|
| | Train | Test | Train–Test | Train | Test | Train–Test |
| Before | $93.02 \pm 0.05$ | $65.75 \pm 0.08$ | 27.27 | $90.60 \pm 0.05$ | $61.64 \pm 0.06$ | 28.96 |
| After | $91.19 \pm 0.09$ | **$67.45 \pm 0.11$** | **23.74** | $93.01 \pm 0.03$ | **$66.36 \pm 0.06$** | **26.65** |

The average accuracy of five repeated tests was used as the final result. Best scores are shown in bold.

### 5. Conclusions

Overfitting is one of the key factors that can affect the performance of DNNs. This paper proposes a new regularization technique called CDM to address the overfitting problem in DNNs. Experimental results demonstrate that CDM consistently enhances network generalization while maintaining or improving network expressiveness, thus preventing overfitting. Furthermore, CDM is easily applicable and can be added to any layer of existing networks.

While our method performs well, there are also some limitations. For example, the combination effect of CDM with other regularization methods is still worth investigating.

**Author Contributions:** T.L., validation, visualization, writing—original draft, writing—review and editing; C.Z., conceptualization, formal analysis, writing—review and editing; X.D., writing—review and editing. All authors have read and agreed to the published version of the manuscript.

**Funding:** This research was funded by the National Natural Science Foundation of China (Project Number 62166016) and the Hainan Provincial Natural Science Foundation of China (Project Number 119MS004).

**Institutional Review Board Statement:** Not applicable.

**Informed Consent Statement:** Not applicable.

**Data Availability Statement:** The paper itself contains all the information required to assess the conclusions. Additional data related to this paper may be requested from the corresponding author.

**Conflicts of Interest:** The authors declare no conflicts of interest.

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
