# Peer review of "A Deep Neural Network Regularization Measure: The Class-Based Decorrelation Method"

_entropy, doi:10.3390/e26010007_

Round 1

Reviewer 1 Report

Comments and Suggestions for Authors

135: Eq.(1) - replace Im by hm

142: Eq.(2) - explain C(.) according (19) in [27]

150: Eq (3) - replace hi by hj in the second term

263: Explain more precisely what "the noise with a standard Gaussian distribution and a weight of ..." means. While the Gaussian variable is a real number, the MNIST pixel has the greyscale 0...255. Does "weight" mean the probability of generating a pixel with noise?

286: Fix "Fig. ??"

Author Response

Response to Reviewer 1 Comments

Comments 1: 135: Eq.(1) - replace Im by hm

Response: Thank you for your valuable feedback! Now, $I_m$ has been corrected to $h_m$. We sincerely apologize for the typographical error. Please kindly check this part at line 135.

Comments 2: 142: Eq.(2) - explain C(.) according (19) in [27]

Response: Thank you for your comments! We are sorry for the unclear presentation for C(.) in our initial submission. Given $\hat{h} = (h_1, h_2, ..., h_m)$ as a set of base classifiers $h_i$ for the sample $(X, Y )$, $C(\dot)$ is any given combination function that minimizes the probatility $P(C(\hat{h}) \neq Y )$. To improve the statement, we have revised the sentences

“the probability of error of the prediction function $p(\cdot)$ is bounded as follows:

\begin{equation}\label{eq2}

       p(C(p(X))\neq Y) \leq \frac{H(Y) - \sum_{i=1}^{m}{I\bigl(h_i(X); Y\bigr)} + D_{LB}(S)}{2},

\end{equation}

which indicates that there is an inverse relationship between the test error and LDiversity.”

to

“Given $\hat{h} = (h_1, h_2, ..., h_m)$ as a set of base classifiers $h_i$ for the sample $(X, Y )$, and $C(\cdot)$ as any given combination function that minimizes the probability $P(C(\hat{h}(X)) \neq Y )$, then the probability is bounded as follows:

\begin{equation}\label{eq2}

       P(C(\hat{h}(X)) \neq Y ) \leq \frac{H(Y) - \sum_{i=1}^{m}{I\bigl(h_i(X); Y\bigr)} + D_{LB}(S)}{2},

\end{equation}

which indicates that there is an inverse relationship between the test error and LDiversity.”

Please kindly check this part from line 141 to line 145 on Page 4 of the revised version.

Comments 3: 150: Eq (3) - replace hi by hj in the second term.

Response: Thank you for your comments! We apologize for the typos that appeared here. We have replaced $h_i$ with $h_j$ and thoroughly reviewed the entire paper to avoid similar issues. Please kindly check this part from line 152 to line 153 on 4 of the revised version.

Comments 4: 263: Explain more precisely what "the noise with a standard Gaussian distribution and a weight of ..." means. While the Gaussian variable is a real number, the MNIST pixel has the greyscale 0...255. Does "weight" mean the probability of generating a pixel with noise?

Response: Thank you for your comments! Prior to introducing noise, we standardize pixel values within the range [0, 1]. Additionally, the term "weights" refers to the coefficient values multiplied with the noise obtained during sampling when added to the original pixel values. Apologies for the confusion caused by the unclear expression here.

The corresponding sentences “Furthermore, to better demonstrate the generalization capabilities of each method, the noise with a standard Gaussian distribution and a weight of 0.2 and 0.05 for MNIST and CIFAR-10 respectively was introduced to the test images” has been modified to “ Moreover, to effectively highlight the generalization capabilities of each method, we introduced noise to the test images. The noise adheres to a standard Gaussian distribution. For MNIST and CIFAR-10 as well as mini-ImageNet, we multiplied the noise obtained by weights of 0.2 and 0.05, respectively, when adding it to the original pixel values. It is important to note that prior to introducing the noise, we standardized pixel values within the range [-1, 1] ”. Please kindly check this part from line 273 to line 278 on Page 8.

Comments 5: 286: Fix "Fig. ??"

Response: Thank you for pointing out this issue. The correct reference is Fig. 3, and we have made the necessary correction. Please kindly check it at line 303 on Page 9.

Reviewer 2 Report

Comments and Suggestions for Authors

The authors propose a novel method for the regularization of artificial neural networks, which promotes variety among neurons belonging to the same layer, through class-based decorrelation.

The methodological insight is well-described and overall easy to catch, although the formalism is quite tedious.

My doubts concern the experimental validation, which is not sufficient from my perspective. MNIST and CIFAR10 are naive datasets, which, nowadays, are not sufficient to assess methodological novelty.

Please, consider more challenging benchmarks to test your method (i.e., Imagenet, ). I would like also to see some real-world case studies (namely, specific domains).

Furthermore, a comparative analysis with the main regularization s.o.t.a. strategies is required.

Minors:

Some references in the introduction are too specific: e.g., [1], which is tailored to corn seed classification. Carefully check all the bibliographies.

Author Response

Response to Reviewer 2 Comments

Comments 1: The authors propose a novel method for the regularization of artificial neural networks, which promotes variety among neurons belonging to the same layer, through class-based decorrelation.
The methodological insight is well-described and overall easy to catch, although the formalism is quite tedious.
My doubts concern the experimental validation, which is not sufficient from my perspective. MNIST and CIFAR10 are naive datasets, which, nowadays, are not sufficient to assess methodological novelty.
Please, consider more challenging benchmarks to test your method (i.e., Imagenet, ). I would like also to see some real-world case studies (namely, specific domains).
Furthermore, a comparative analysis with the main regularization s.o.t.a. strategies is required.

Response: Thank you for your comments! In response to your suggestions, we have conducted new experiments on the ImageNet dataset. However, the extensive data available in ImageNet plays a pivotal role in mitigating the risk of overfitting during model training on this dataset. To intentionally create a scenario more prone to overfitting, our focus was specifically on learning tasks conducted on the mini-ImageNet dataset, in contrast to the larger ImageNet dataset. In mini-ImageNet, each of the 100 classes is limited to only 500 training samples, heightening the susceptibility of models with deep structures trained on this dataset to exhibit overfitting phenomena. Furthermore, we adopted two state-of-the-art (SOTA) methods, i.e., InceptionV3 and MobileNet, pretrained on the ImageNet dataset as baseline approaches. Both of these approaches employ a variety of effective regularization techniques, including established strategies such as dropout, batch normalization, and L2 regularization. Additionally, these methods incorporate innovative regularization techniques like global average pooling and controlling the size of channels.

We applied CDM to both of the SOTA methods, comparing their performance variations before and after the application of CDM. The results revealed an enhancement in test accuracy and a reduction in generalization error following the implementation of CDM. Concretely, the experiments demonstrate that the methods incorporating CDM result in improved classification accuracy for all methods compared to the original approach, with increases of 1.7 and 4.72 points, respectively. Concurrently, the integration of CDM reduces generalization errors by 3.53 and 2.31 points, respectively. The comparison with the SOTA methods strongly underscores the effectiveness of CDM regularization.

Please kindly check this part at line 270 and line 290 on page 8, also the content from line 368 to line 382 as well as the Table 4 on Page 13.

Comments 2: Minors:
Some references in the introduction are too specific: e.g., [1], which is tailored to corn seed classification. Carefully check all the bibliographies.

Response: Thank you for your feedback. We have removed reference [1] from the original text and thoroughly reviewed all associated references. References [2-4], which presented similar issues in the original text, have been eliminated. Furthermore, we have included "LeCun, Y.; Bengio, Y.; Hinton, G. Deep Learning. Nature 2015, 521, 436–444" as the primary reference [1] in the revised version. Please kindly check this part from line 12 to line 18 on Page 1.

Round 2

Reviewer 2 Report

Comments and Suggestions for Authors

The authors answered all of my suggestions. I recommend to accept the paper.